# Stability of Dengue 2 Nonstructural Glycoprotein 1 (NS1) Is Affected by the Nature of Basic Residue at Position NS1-324

Eva Ogire, Chaker El-Kalamouni, Philippe Desprès and Marjolaine Roche *

Unité Mixte de Recherche Processus Infectieux en Milieu Insulaire Tropical (PIMIT),
Université de La Réunion, INSERM U1187, CNRS 9192, IRD 249, Plateforme Technologique CYROI,
94791 Sainte Clotilde, La Réunion, France
* Correspondence: marjolaine.roche@univ-reunion.fr

**Abstract:** Dengue is the most prevalent mosquito-borne viral disease. It is caused by the infection of any of the four dengue virus (DENV) serotypes DENV-1 to DENV-4. The DENV non-structural glycoprotein 1 (NS1) plays an important role in virus replication and the immunopathogenesis of virus infection. The NS1 protein has been identified as both a cell-associated homodimer and a soluble secreted lipoprotein nanoparticle. The nature of the residues at positions NS1-272 and NS1-324 in the β-ladder domain may have an effect on the biological behaviors of DENV-2 NS1 protein in human hepatoma Huh7 cells. The stability of the NS1 protein from the Reunion 2018 DENV-2 strain was affected by the presence of lysine residues at positions 272 and 324. In the present study, we evaluated the impact of mutations into lysine at positions 272 and 324 on recombinant NS1 protein from the DES-14 DENV-2 strain bearing arginine residue on these two positions. The DES-14 NS1 protein mutant bearing a lysine at position 324 was deficient in protein stability and secretion compared to wild-type protein. The defect in the DES-14 NS1 protein mutant was associated to oxidative stress and pro-inflammatory cytokine activation in Huh7 cells. The ubiquitin-proteasome proteolytic pathway might play a key role in the stability of DENV-2 protein bearing a lysine residue at position 324.

**Keywords:** dengue virus; non-structural protein 1; recombinant viral protein; Huh7 cells; lysine residue; ubiquitin-proteasome system; oxidative stress; inflammation

## 1. Introduction

Dengue, which became one of the most prevalent mosquito-borne viral diseases in tropical areas worldwide [1], is caused by four distinct serotypes of dengue virus (DENV-1, DENV-2, DENV-3, and DENV-4) after the bite of an infected *Aedes* species mosquito. Infected individuals who became symptomatic can develop a wide spectrum of clinical manifestations ranging from a flu-like disease (dengue fever) to severe illness (severe dengue), potentially leading to lethal complications [2,3]. To date, there is no specific treatment for severe dengue. In recent years, two tetravalent dengue vaccines, Dengvaxia (a registered trademark of Sanofi) and QDENGA (a registered trademark of Takeda Vaccines), have been approved for use in several countries where dengue is endemic [4,5].

The incidence of dengue has recently grown in the southwest Indian Ocean (SWIO), notably in French territory, La Reunion Island, with the emergence of a cosmopolitan genotype of DENV-2 in 2017, followed by DENV-1 in 2020 [6,7]. There were thousands of infections and dozens of deaths associated with severe dengue in La Reunion Island in 2021. We recently reported that the non-structural glycoprotein 1 (NS1, 352 amino-acid residues) from epidemic DENV-2 strain RUN-18 isolated from human infection in La Reunion in 2018 was weakly expressed in human hepatoma Huh7 cells in comparison to the NS1 glycoprotein from epidemic DENV-2 strain DES-14 isolated from human infection in Tanzania in 2014 [8]. The instability of RUN-18 NS1 mostly relates to two lysine residues at positions 272 and 324 in the β-ladder domain of the protein.

The impact of protein mutations on the pathogen expression has been well documented [9–11]. A limited number of amino acid substitutions may have a major effect on infection ability and host–pathogen interactions [12–16]. Consequently, it is critically important to investigate the consequences of protein mutations on the biological behaviors of the pathogen. In the present study, we wondered whether each of the two lysine residues at positions 272 and 324 contributes to the instability of RUN-18 NS1 protein in Huh7 cells [8]. For this purpose, we used a recombinant NS1 glycoprotein from the Cosmopolitan DENV-2 strain D2-K2_RIJ059/Dar es Salaam 2014 (hereinafter called DES-14) [8,17]. DES-14 NS1 differentiates from RUN-18 NS1 by the presence of two arginine residues on positions 272 and 324. We reported that recombinant DES-14 NS1 is correctly expressed in Huh7 cells leading to efficient secretion of soluble protein [8]. We showed that amino-acid changes Arg-to-Lys on position 324, and to a lesser extent on position 272, may lead to a decrease in DENV-2 NS1 protein stability.

## 2. Materials and Methods

### 2.1. Cell Lines and Antibodies

Human hepatoma Huh7 cells (a generous gift from Dr V. Lotteau, CIRI, Lyon) were cultured in DMEM medium supplemented with 10% heat-inactivated fetal bovine serum (Dutscher, Brumath, France) and antibiotics (PAN Biotech Dutscher, Brumath, France) at 37 °C under a 5% $CO_2$ atmosphere. The mouse anti-DDDDK tag (FLAG) antibody (Abcam, Cambridge, UK) in PBS, rabbit DENV-2 NS1 polyclonal antibody PA5-33207 (anti-NS1 pAb), and goat Alexa 488-conjugated anti-mouse IgG antibody were purchased from Thermo Fisher Scientific (Les Ulis, France) in PBS. The mouse anti-flavivirus NS1 antibody [D/2/D6/B7] (anti-NS1 mAb) and anti-mouse IgG HRP-conjugated secondary antibodies were purchased from Abcam (Cambridge, UK). The rabbit anti-β actin polyclonal antibody was purchased from ABclonal (MA, Woburn, USA).

### 2.2. Vector Plasmids Expressing Recombinant DENV-2 NS1 Proteins

A synthetic mammalian codon-optimized gene coding for the authentic NS1 signal peptide followed by the residues NS1-1/352 of the Cosmopolitan DENV-2 strain D2-K2_RIJ059/Dar es Salaam 2014 (Genbank accession number MG189962) was synthetized by Genecust (Boynes, France). A Kozak consensus sequence for initiation of translation was inserted at the 5′ end of the synthetic gene. The cloning of the synthetic gene into pcDNA3.1 plasmid and the sequencing of the resulting plasmid pcDNA3/DES-14.NS1 were performed by Genecust (Boynes, France). Direct mutagenesis on pcDNA3/DES-14.NS1 to generate pcDNA3/DES-14.NS1-(K272), pcDNA3/DES-14.NS1-(K324), or pcDNA3/DES-14.NS1-(K272, K324) was performed by Genecust (Boynes, France). The production of endotoxin-free plasmids, the DNA quantification, and their sequencing were performed by Genecust (Boynes, France). Cells were transfected with endotoxin-free plasmids (2.5 μg DNA per $10^6$ cells) using Lipofectamine 3000 (Thermo Fisher Scientific, Les Ulis, France) according to the manufacturer's instructions.

### 2.3. Immunoblot Assay

Huh7 cell lysates were performed in RIPA lysis buffer (Sigma, France). All subsequent steps of immunoblotting were performed as described [8]. Membranes were incubated with a primary antibody at dilution 1:200 and then with an anti-mouse or anti-rabbit IgG HRP-conjugated secondary antibody at 1:5000 dilution. Pre-stained natural protein standards for SDS-PAGE were used to estimate the apparent molecular weights of intracellular proteins. Membranes were developed with Pierce ECL Western blotting substrate (Thermo Fisher Scientific, France) and exposed on an Amersham imager 680 (GE Healthcare, Buc, France). The signal intensity of intracellular and soluble forms of rNS1 was measured by Image J software. The results are the mean of at least three independent assays. The protein abundance of β-actin was determined and was equivalent for each intracellular form.

## 2.4. Proteasome Inhibition Assay

Huh7 cell monolayers in 12-well culture plate were transfected with plasmids expressing recombinant DENV-2 NS1 protein. At 18 h post-transfection, cells were incubated for 6 h with 10 μM MG132 in 0.1% DMSO or 0.1% DMSO as vehicle control. Effects of proteasome inhibitor MG132 on NS1 expression were analyzed by immunoblot assay.

## 2.5. RT-qPCR

Huh7 cells were seeded in 12-well culture plate and transfected the following day. Then, 24 h after transfection, total RNA was extracted from cells with RNeasy kit (Qiagen, Venlo, Netherlands) according to the manufacturer's instructions. Reverse Transcription was performing using M-MLV Reverse Transcriptase (Life Technologies, Les Ulis, France). Quantitative PCR was performed on CFX Connect Real-Time PCR system (Biorad Laboratories, Marnes-La-Coquette, France). Briefly, 10 ng cDNA was amplified using 0.2 μM of each primer in SYBR Green real-time PCR master mix buffer (1X) (Thermo Fisher Scientific, Les Ulis, France). The qPCR data sets were analyzed using the $\Delta\Delta Ct$ method and the results were normalized to Human Glyceraldehyde 3-phosphate dehydrogenase (GAPDH), which was used as internal control. The primers used in this study are listed in Table S1.

## 2.6. Measurement of ROS

Huh7 cells were seeded in 96-well culture plate and transfected the following day. Then, 24 h after transfection, intracellular reactive oxygen species (ROS) level was measured using the CM-$H_2$DCFDA dye. For total ROS detection, cells were stained with 10 μmol·$L^{-1}$ CM-$H_2$DCFDA in PBS buffer for 45 min at 37 °C in the dark. ROS level was quantified by fluorescence at 520 nm using a FLUOstar Omega Microplate Reader (BMG Labtech, Champigny sur Marne, France).

## 2.7. Statistical Analysis

An unpaired *t*-test was used to compare quantitative data. GraphPad Prism 9.5.0 for macOS (www.graphpad.com) was used for all statistical analysis and resulting graphs.

## 3. Results

### 3.1. Effect of Amino-Acid Substitutions R272K and R324K on DES-14 rNS1 Protein

We reported that the nature of basic residues at positions 272 and 324 influences the behavior of DENV-2 NS1 protein in Huh7 cells [8]. The presence of Lys residues at positions 272 and 324 affected the stability of RUN-18 NS1 protein [8]. To better understand the respective role of Lys272 and Lys324 in RUN-18 NS1 protein stability, we evaluated the impact of point mutations into Lys on DES-14 NS1 protein bearing Arg residues on these two positions. By site-directed mutagenesis, amino-acid substitutions R272K or/and R324K were introduced into recombinant DES-14 NS1 protein (rNS1$^{wt}$) in order to generate three NS1 mutants, namely rNS1-(K272), rNS1-(K324), and rNS1-(272, K324), respectively. To determine whether structural information can predict the impact of mutations on DENV-2 NS1 protein structure, 3D structure prediction of DES-14 NS1 protein and its mutants was performed by modeling on PHYRE$^2$ protein fold recognition server (Figure 1). Structural analysis of NS1 showed that the two-point mutations Arg-to-Lys at positions 272 and 324 have no obvious effect on protein conformation.

We first evaluated the expression level of recombinant DES-14 rNS1 and its DES-14 rNS1-(K272, K324) mutant in Huh7 cells using recombinant plasmids pcDNA3/DENV-2.rNS1$^{DES-14}$ or pcDNA3/DENV-2.rNS1$^{DES-14}$-(K272, K324) expressing FLAG-tagged rNS1 proteins [8]. By immunofluorescence assay, using anti-FLAG antibody on Huh7 cells transfected for 24 h with pcDNA3/DENV-2.rNS1$^{DES-14}$ or pcDNA3/DENV-2. rNS1$^{DES-14}$-(K272, K324), we found a similar transfection efficacy regardless the NS1 construct (Figure S1). Such results confirm that the plasmids expressing DES-14 rNS1 and the mutants carrying the lysine at positions 272 and 324 are suitable for the analysis of the NS1 protein. For

the further experiments, we used plasmids expressing recombinant DES-14 NS1 proteins without any C-terminal tag.

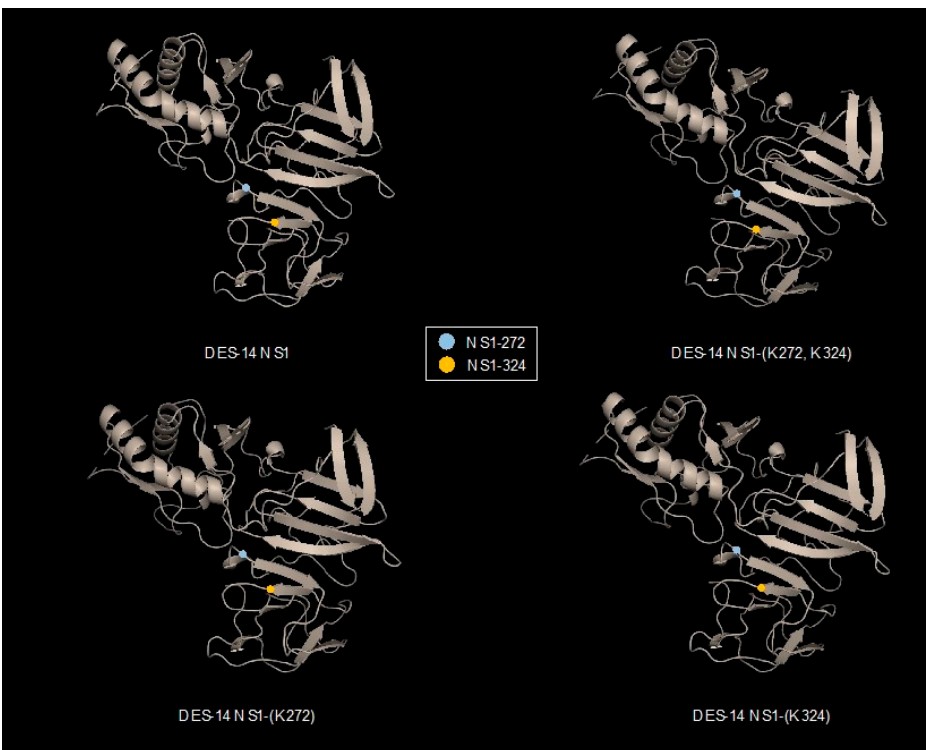

**Figure 1.** Structural features of DENV-2 NS1 protein. Tridimensional structure prediction of DES-14 NS1 protein (Genbank access number MG189962) and the three mutants DES-14 NS1-(K272, K324), NS1-(K272), and NS1-(K324) was performed by modeling on PHYRE² protein recognition server (http://www.sbg.bio.ic.ac.uk/~phyre2/html/page.cgi?id=index; accessed on 23 December 2022). The predicted structures were analyzed with PyMOL 2.5, a program for interactive visualization of tridimensional proteins. The colored dots indicate the positions of residues NS1-272 and NS1-324 in the 3D structure of NS1 protein.

Huh7 cells were transfected for 24 h with pcDNA3/DES-14.NS1 plasmids expressing rNS1^wt or its mutants bearing lysine residues at positions 272 and/or 324 (Figure 2). Cell supernatants and RIPA cell lysates were assayed for rNS1 expression by immunoblot assay using mouse monoclonal anti-flavivirus antibody NS1 D/2/D6/B7 (anti-NS1 mAb) that recognizes different oligomeric forms of NS1. Expression of rNS1^wt and its mutants were compared by immunoblot assay using anti-NS1 mAb (Figure 2A). As a positive control, we observed that the amount of rNS1-(K272, K324) mutant was lower in Huh7 cells compared to rNS1^wt. Analysis of rNS1 mutants revealed that the substitutions R324K and, to a lesser extent, R272K may have an influence on rNS1 expression level. The determination of signal intensity of immunolabeled rNS1 revealed that the amino-acid substitution R272K reduced the protein expression level by 50% reaching 90% with the amino-acid substitution R324K in comparison to rNS1^wt (Figure 2B). These results showed that a lysine residue on position NS1-272 or NS1-324 influenced the DENV-2 rNS1 expression level. Analysis of culture supernatants from Huh7 cells expressing rNS1 by a dot-blot assay revealed that the efficacy of rNS1 release was mostly affected by the amino-acid substitution R324K (Figure 2A). We found that the amount of secreted soluble rNS1 mutant bearing the Lys324 residue represented only 20% of rNS1^wt (Figure 2B). Whereas amino-acid substitution R272K has a minor effect on the secretion of rNS1^wt by Huh7 cells. These results highlighted a major effect of Lys residue at position 324 on the biological behavior of DES-14 rNS1 protein.

**A.**

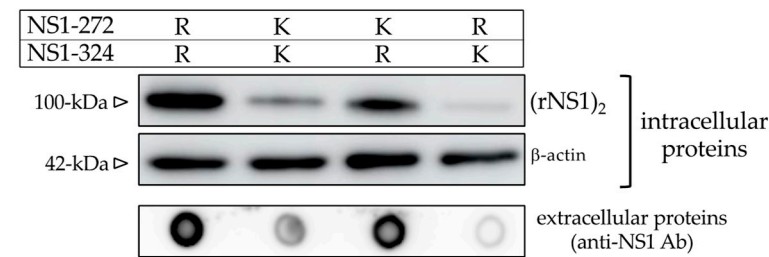

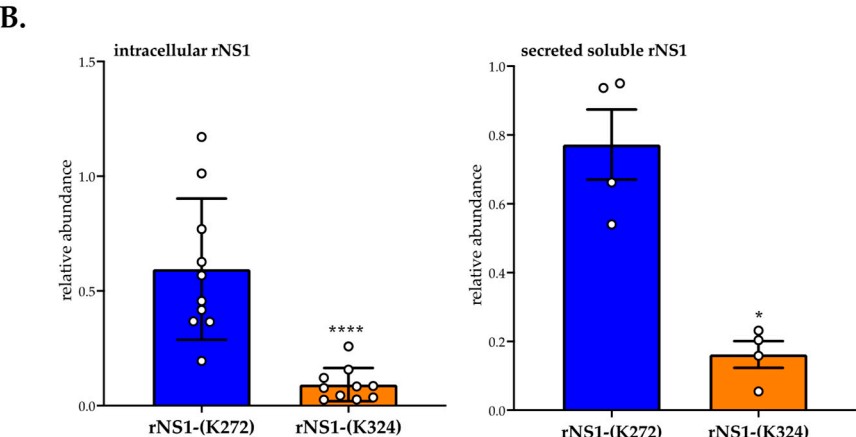

**B.**

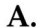

**Figure 2.** Effects of amino-acid substitutions R272K and R324K on DENV-2 rNS1 expression. Huh7 cells were transfected for 24 h with plasmids expressing DENV-2 rNS1$^{wt}$ or rNS1-(K272), rNS1-(K324), and rNS1-(K272, K324) mutants. In (**A**), immunoblot assay was performed on non-heated RIPA cell lysates (intracellular proteins) and cell supernatants (extracellular proteins) using anti-NS1 antibody D/2/D6/B7 (anti-NS1 Ab). The intracellular NS1 dimer is indicated as (rNS1)$_2$. The β-actin protein served as a protein loading control. The estimated apparent molecular weights of (rNS1)$_2$ and β-actin are indicated. The basic residues at the positions NS1-272 and NS1-324 are shown at the top. In (**B**), signal intensity was quantified using ImageJ software to evaluate the amount of rNS1 mutants relative to rNS1$^{wt}$. The results are the mean ($\pm$ SEM) of 10 (intracellular rNS1) or 4 (secreted rNS1) replicates (open circles). *p*-values were determined on the comparison of rNS1-(K324) with rNS1-(K272) mutant (**** $p < 0.0001$; * $p < 0.05$).

### 3.2. Effect of Proteasome Inhibition on rNS1 Protein Expression in Huh7 Cells

We wondered whether the Lys324 residue could act as a potential site of ubiquitin ligation which targets rNS1-(Lys324) mutant for proteasome-mediated degradation leading to a defect in rNS1 accumulation in Huh7 cells [18]. Consequently, Huh7 cells were transfected with plasmids expressing rNS1$^{wt}$ or its mutants and, 18 h post-transfection, 10 μM proteasome inhibitor MG132 was added for 6 h. An immunoblot assay using anti-NS1 mAb and a rabbit DENV-2 NS1 polyclonal PA5-32207 (anti-NS1 pAb) that recognizes linear epitopes in the NS1 region 117–301 showed that MG132 influence rNS1 expression level (Figure 3A). At the dose of 10 μM, proteasome inhibitor significantly increased the protein expression level of rNS1-(K324) mutant but not rNS1-(K272) as compared to rNS1$^{wt}$ (Figure 3B). There was a two-fold increase in the amounts of different oligomeric forms of rNS1-(K324). The ability of proteasome inhibitor MG132 to increase rNS1-(K324) mutant expression level supports a role for proteasome-mediated degradation in the instability of protein in Huh7 cells.

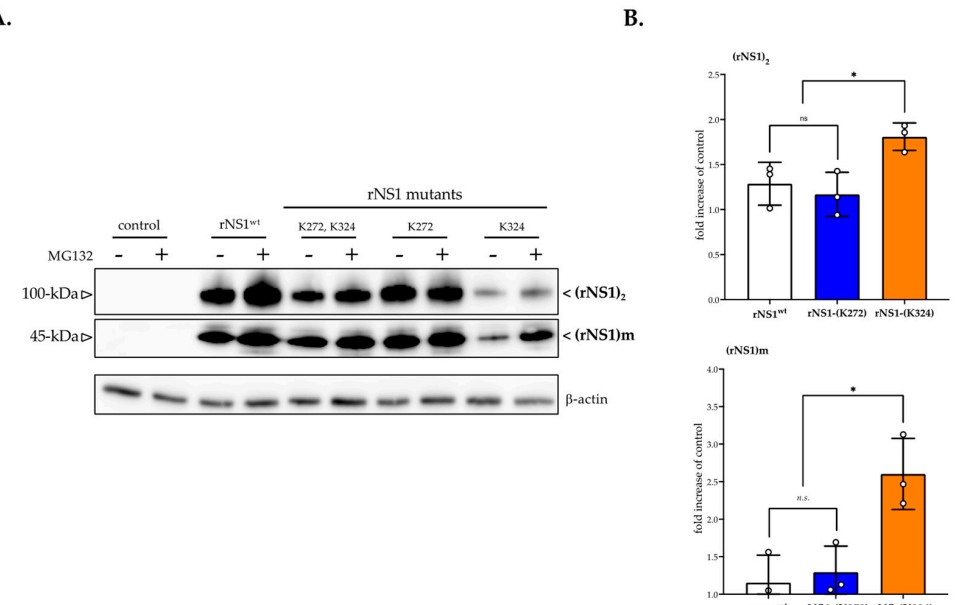

**Figure 3.** Effect of proteasome inhibition on DENV-2 rNS1 expression. Huh7 cells were transfected for 24 h with plasmids expressing DENV-2 rNS1$^{wt}$ or rNS1-(K272), rNS1-(K324), and rNS1-(K272, K324) mutants or mock-transfected (control) in presence (+) or absence of 10 μM of the proteasome inhibitor MG132 in 0.1% DMSO for 6 h. In (**A**), immunoblot assays were performed on RIPA cell lysates using anti-NS1 antibody. Cell lysates were analyzed before (top) and after (bottom) heat-denaturation in the presence of a reducing agent. The mouse anti-NS1 antibody D/2/D6/B7 (anti-NS1 mAb) was used to detect heat-labile dimer rNS1 which is indicated as (NS1)$_2$. The rabbit anti-NS1 antibody PA5-33207 (anti-NS1 pAb) was used to detect monomeric form of rNS1 which is indicated as (rNS1)m. The β-actin protein served as a protein loading control. The estimated apparent molecular weights of (rNS1)$_2$ and (rNS1)m are indicated. In (**B**), signal intensity was quantified using ImageJ software to calculate the fold-increase between the amounts of rNS1 expressed with or without MG132 (control). The results are the mean ($\pm$ SEM) of 3 replicates (open circles). *p*-values were determined on the comparison of different rNS1 (* *p* < 0.05; *n.s.*: not significant).

### 3.3. Ability of rNS1-(Lys324) Mutant to Trigger Oxidative Stress and Pro-Inflammatory Cytokine Transcription Activation

The above results suggest that ubiquitin-proteasome system (UPS) is involved in the low protein expression level of rNS1-(Lys324) mutant in Huh7 cells. Thus, the Lys324 residue could be recognized as a potential site of ubiquitin ligation which targets DENV-2 NS1 protein for proteasome-mediated degradation. We investigated whether proteasomal degradation of rNS1-(Lys324) mutant is associated to a greater accumulation of reactive oxygen species (ROS) that are potentially capable of interfering with cellular processes. The fluorescent probe dichlorodihydrofluorescein diacetate (DCFH-DA) that measures total intracellular ROS has been used to quantitatively compare relative ROS levels in Huh 7 cells transfected for 24 h with different plasmids expressing rNS1 (Figure 4). Intracellular ROS are known to be highly unstable; thus, a very low concentration of ROS can have a major effect on cellular metabolism. There was a significant increase in DCFH-DA signal in Huh 7 cells expressing rNS1-(K272) or rNS1-(K324) mutants as compared to rNS1$^{wt}$. However, expression of rNS1-(K324) mutant resulted to a greater accumulation of intracellular ROS as compared to rNS1-(K272) mutant (Figure 4A). The *SOD1* gene encodes superoxide dismutase-1 (SOD1) that acts as antioxidant enzyme protecting the cells from ROS. RT-qPCR analysis on SOD1 mRNA was performed on Huh 7 cells expressing rNS1. Expression of rNS1-(K324) mutant but not rNS1-(K272) mutant caused a significant increase in the amount of SOD1 gene transcripts in Huh7 cells as compared to rNS1$^{wt}$ (Figure 4B). These results suggest that rNS1-(K324) mutant is an efficient inducer of intracellular ROS in Huh7 cells.

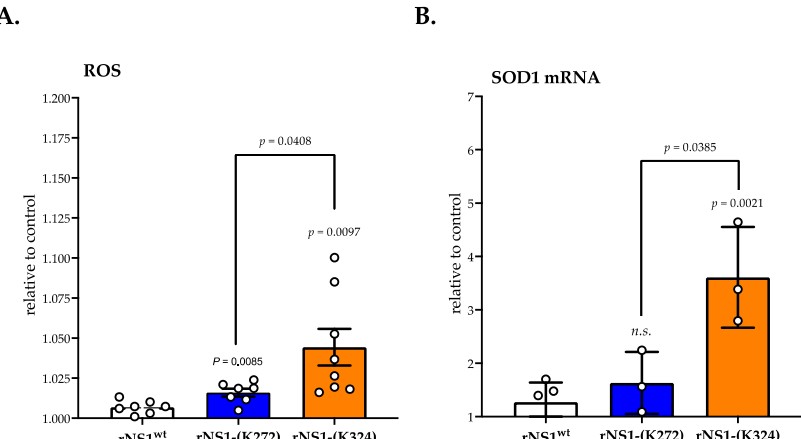

**Figure 4.** ROS production in Huh7 cells expressing the rNS1-(K24) mutant. Huh7 cells were transfected for 24 h with plasmids expressing DENV-2 rNS1$^{wt}$, rNS1-(K272), or rNS1-(K324). In (**A**), the production of total intracellular reactive oxygen species (ROS) was measured using DCFH-DA dye. Intracellular ROS in samples was expressed as a fold change relative to mock-transfected cells. In (**B**), antioxidant enzyme superoxide dismutase (SOD-1) mRNA levels were quantified by RT-qPCR. Results are expressed as the fold induction of transcripts in samples relative to those in mock-transfected cells. The results are the mean ($\pm$ SEM) at least seven (**A**) and three (**B**) replicates (open circles). *p*-values were determined on the comparison of rNS1 mutants versus rNS1$^{wt}$ and rNS1-(K324) mutant versus rNS1-(K272) mutant (*n.s.*: not significant).

We wondered whether the production of intracellular ROS causes up-regulation of pro-inflammatory cytokines IL-1β and IL-6 in Huh7 cells expressing DENV-2 rNS1. RT-qPCR was performed on Huh7 cells transfected for 24 h with plasmids expressing rNS1$^{wt}$ or rNS1-(K272) and rNS1-(K324) mutants (Figure 5). There was a weak but significant increase in the IL-1β transcript numbers in Huh7 cells expressing rNS1-(K324) mutant (Figure 5A). An up-regulation of IL-6 mRNA expression was also observed in Huh7 cells (Figure 5B). The expression of rNS1-(K272) and rNS1-(K324) mutants stimulated a higher transcription level of IL-6 compared to rNS1$^{wt}$ (Figure 5). Taken together, these results suggest that expression of rNS1-(K324) mutant and, to a lesser extent, rNS1-(K272) mutant can generate intracellular ROS associated to the activation of pro-inflammatory cytokines in Huh7 cells.

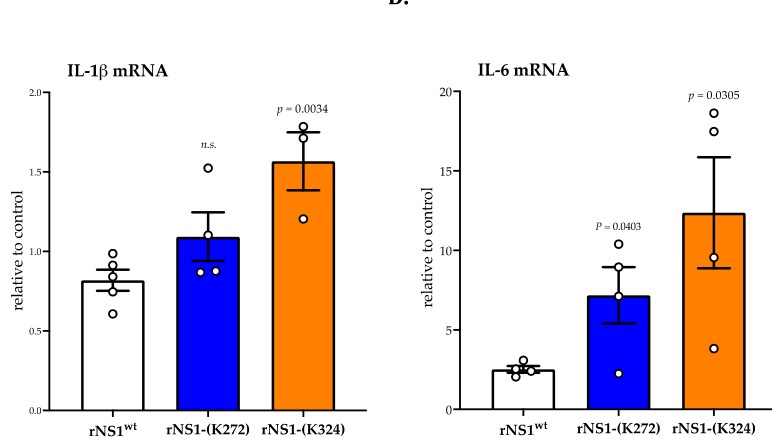

**Figure 5.** Pro-inflammatory cytokine activation in Huh7 cells expressing the rNS1-(K324) mutant. Huh7 cells were transfected for 24 h with plasmids expressing rNS1$^{wt}$ or the rNS1-(K272) and rNS1-(K324) mutants. The IL-1β (**A**) and IL-6 (**B**) transcripts were quantified by RT-qPCR. Results are expressed as the fold induction of transcripts in samples relative to those in mock-transfected cells. The results are the mean ($\pm$ SEM) at least three replicates (open circles). *p*-values were determined on the comparison of rNS1 mutants versus rNS1$^{wt}$.

## 4. Concluding Remarks

DENV NS1 glycoprotein has been identified as essential for virus replication and viral pathogenesis. The intracellular NS1 present in the ER lumen exists as a hydrophobic dimer associated with the replication complexes and the protein is also found on cell membranes [19–21]. The soluble form of NS1 has been identified as a hydrophobic lipoprotein nanoparticle that is excreted in the extracellular compartment [22,23]. The soluble DENV NS1 has been implicated in a variety of processes which are thought to contribute to dengue pathogenesis such as immune evasion and endothelial dysfunction [24–26].

Analysis of recombinant NS1 glycoprotein from clinical isolate RUN-18 that was collected during the DENV-2 epidemic in La Reunion in 2018 showed that the protein was poorly stable in human hepatoma Huh7 cells [8]. In contrast, a high expression level of recombinant NS1 protein from clinical isolate DES-14 that was collected during the DENV-2 epidemic in Tanzania in 2014 was observed in Huh7 cells. Notably, DES-14 NS1 protein contains Arg residues on positions 272 and 324. In order to gain more insight into the impact of Lys272 and Lys324 on DENV-2 NS1 protein stability, the arginine residues of DES-14 NS1 protein at positions 272 and 324 were substituted by site directed mutagenesis to lysine. Expression analysis of recombinant DES-14 NS1-(K272, K324), as well as DES-14 NS1-(K272) and DES-14 NS1-(K324) mutants, highlighted an essential role for Lys324 residue in the instability of DENV-2 NS1 in Huh7 cells.

We wondered whether the instability of the DES-14 NS1-(K324) mutant is the consequence of a proteasome-mediated protein degradation of protein in Huh7 cells [27]. Indeed, the C-terminal lysine residues of NS1 could be targeted by UPS [18]. The ability of proteasome inhibitor MG132 to increase the protein level of DES-14 NS1-(K324) mutant in Huh7 cells suggests that Lys324 could engage DENV-2 NS1 protein in a proteasome-mediated degradation. It is well documented that activation of the ubiquitin-proteasome system may cause ER-stress associated to an increase in ROS production [28]. Consistent with this finding, expression of the DES-14 NS1-(K324) mutant resulted in increased production of ROS associated to the transcriptional activation of the antioxidant SOD1 enzyme. Given that increase in ROS production can trigger activation of a pro-inflammatory response as a host-cell defense mechanism [29,30], the up-regulation of mRNA cytokines IL-1β and IL-6 was observed in Huh 7 cells overexpressing the DES-14 NS1-(K324) protein mutant. Thus, the presence of a lysine at position DENV-2 NS1-324 would have a propensity to promote the proteasome-mediated degradation of protein, triggering oxidative stress and pro-inflammatory cytokine activation in human hepatoma cells (Figure 6).

Although arginine and lysine are two basic residues, only the latter is considered as a substrate for protein ubiquitination, SUMOylation, or acetylation [31–33]. In the present study, we revealed the effects of a lysine residue at position 324 on the post-translational processing of DENV-2 NS1 protein. Sequence analysis of different DENV-2 NS1 proteins revealed that Arg residue was predominant at position NS1-324 (Table S2). Among few DENV-2 NS1 proteins bearing a lysine on position 324, the presence of Arg residue on position 272 has not been observed so far. Notably, the NS1 sequence of the Reunion 18 DENV-2 strain includes a lysine on both positions 272 and 324. SUMOylation is a post-translational process involved in DENV replication [34]. A contribution of NS1-Lys272 residue in a putative SUMO-interactive motif G*K*LE has been hypothesized [8]. The possible involvement of Lys272 in a mechanism of the SUMOylation process could confer a relative resistance of Reunion 2018 DENV-2 NS1 protein to proteasome-mediated degradation. Therefore, it is a priority to evaluate the impact of basic residues at positions NS1-272 and NS1-324 on DENV-2 growth in Huh7 cells. Improved knowledge by which lysine residues at position NS1-272 and NS1-324 influence the behavior of epidemic Reunion 18 DENV-2 strain will broaden our understanding on the role of NS1 protein in the immunopathogenesis of dengue disease.

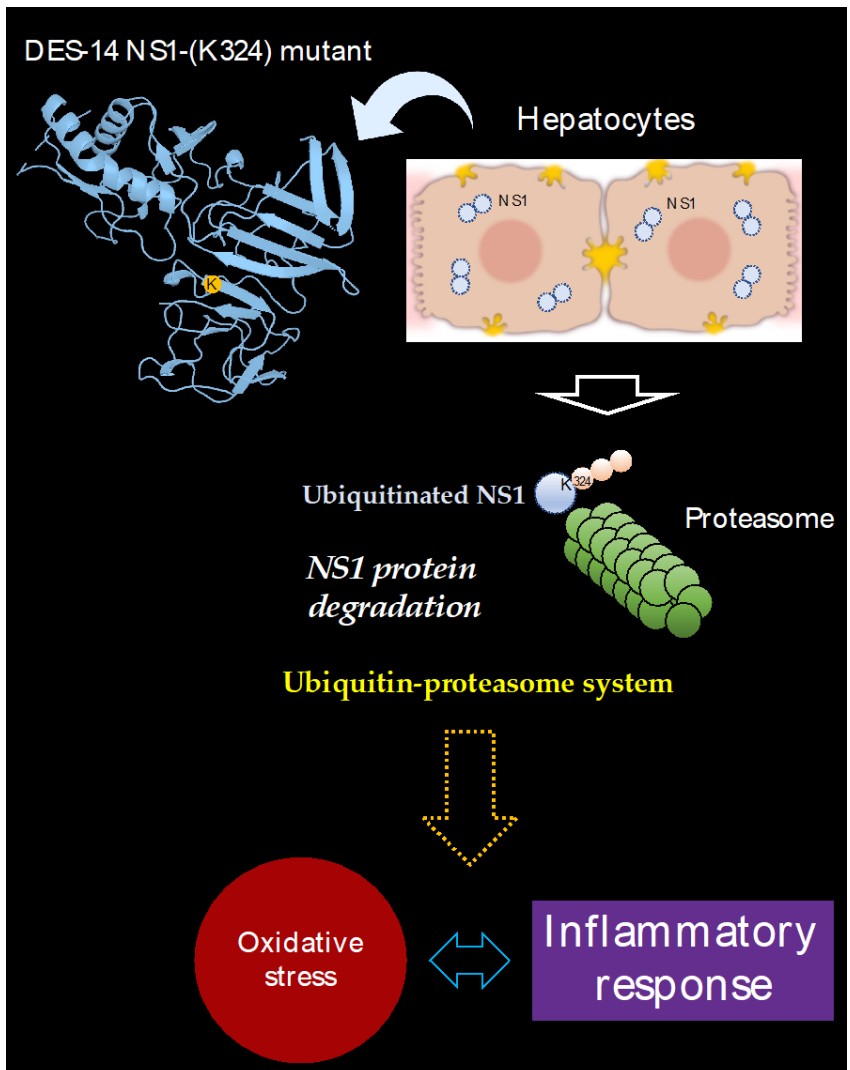

**Figure 6.** Expression of DES-14 NS1 protein mutant bearing a lysine residue at position 324 (DES-14 NS1-(K324) mutant) is associated to oxidative stress and inflammatory in human hepatoma Huh7 cells.

**Supplementary Materials:** The following supporting information can be downloaded at: https://www.mdpi.com/article/10.3390/cimb45020106/s1, Table S1: List of primers for RT-qPCR analysis. Table S2: Amino-acid residues at positions NS1-272 and NS1-324 among DENV-2 strains. Figure S1: Expression of recombinant DES-14 NS1 proteins in Huh7 cells.

**Author Contributions:** E.O., P.D. and M.R. conceived and designed the experiments; E.O. performed the experiments; E.O., P.D. and M.R. analyzed the data; E.O., C.E.-K., P.D. and M.R. contributed to reagents/materials/analysis tools; E.O., P.D., and M.R. wrote the paper. All authors have read and agreed to the published version of the manuscript.

**Funding:** This work was supported by the European Regional Development Fund (ERDF) through the RUNDENG project (N° 0202640-0022937) and the POE FEDR 2014-20 of the Conseil regional de La Réunion (PHYTODENGUE program, N° SYNERGIE: RE0028005).

**Institutional Review Board Statement:** Not applicable.

**Data Availability Statement:** The data and materials generated during the study are available in the PIMIT lab.

**Acknowledgments:** We greatly thank P. Mavingui for support. The authors thank all the members of MOCA team and P.O. Vidalain, O. Diaz, and V. Lotteau for their help and useful discussion.

**Conflicts of Interest:** The authors declare no conflict of interest.

## Abbreviations

| | |
|---|---|
| Ab | Antibody |
| DCFH-DA | Dichlorodihydrofluorescein diacetate |
| DENV | Dengue virus |
| DES-14 | DENV-2 strain D2-K2_RIJ059/Dar es Salaam 2014 |
| DMEM | Dulbecco's Modified Eagle Medium |
| ECL | Enhanced chemiluminescent |
| HRP | Horseradish-peroxidase |
| Il-1β | Interleukine 1β |
| IL-6 | Interleukine 6 |
| kDa | kiloDalton |
| mAb | Monoclonal antibody |
| mRNA | Messager Ribonucleic acid |
| NS1 | Non-structural protein 1 |
| pAb | Polyclonal antibody |
| rNS1 | Recombinant non-structural protein 1 |
| ROS | Reactive Oxidative Species |
| RT-qPCR | Reverse Transcription- quantitative Polymerase Chain Reaction |
| RUN-18 | DENV-2 strain RUN-18 (or RUJul, Genbank accession number MN272404) |
| SDS-PAGE | Sodium dodecyl sulfate-polyacrylamide gel electrophoresis |
| SOD1 | Superoxide dismutase 1 |
| SWIO | Southwestern Indian Ocean |
| UPS | Ubiquitin-proteasome system |

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
