# Peer review of "Stability of Dengue 2 Nonstructural Glycoprotein 1 (NS1) Is Affected by the Nature of Basic Residue at Position NS1-324"

_cimb, doi:10.3390/cimb45020106_

Round 1
Reviewer 1 Report
Dear Autors,
The topic is really interesting to Science. Mainly it is a priority to evaluate new alternatives in immunopathogenesis of dengue disease. However, I consider that the conclusions should be focus in the main results obtained. Please, to consider no references in the conclusions.
Thanks
Author Response
December 30th, 2022
To Current Issues in Molecular Biology (CIMB)
Section Biochemistry, Molecular and Cellular Biology
REF: Revised version of MS#cimb-2107494-R1 (Communication)
Dear Editors,
Dear Reviewers,
Thank you for your high consideration of our submitted manuscript.
Please in the revised document our point-to-point responses to Reviewers #1, #2, and #3.
Two major changes made to the article can be found below.
#1. In the revised version of the manuscript, we provided new Figures 1 and 6. A tridimensional structure prediction was performed on DENV-2 NS1 protein (Figure 1). A synthetic scheme on the biological behavior of DENV-2 NS1 mutant in Huh7 cells has been proposed as Figure 6.
#1. In the revised version of the manuscript, we provided new Figure S1. Immunofluorescence assay was performed on Huh7 cells expressing a recombinant DENV-2 NS1 protein.
We greatly appreciate the efforts of the three reviewers and the editorial team of CIMB to help us make this work the strongest possible, and we look forward to your reply.
Thank you very much for your assistance and we look forward to your earliest reply.
Sincerely yours,
Dr Marjolaine ROCHE
Corresponding author
Dr Marjolaine ROCHE, PhD
Faculty of Health Biology, La Reunion University
UM 134 PIMIT
CYROI platform, 97490 Sainte-Clotilde, La Réunion. France.
e.mail: marjolaine.roche@univ-reunion.fr
We greatly thank Reviewer #1 for his/her thoughtful comments and the opportunity to revise our manuscript.
Responses to Reviewer #1:
Q1. The topic is really interesting to science. Mainly it is a priority to evaluate new alternatives in immunopathogenesis of dengue disease. However, I consider that the conclusions should be focus in the main results obtained. Please, to consider no references in the conclusions.
We appreciate the remark by the reviewer. According to the reviewer’s comment, the text has been modified in Concluding remarks paragraph of the revised manuscript. We emphasize on the effects of lysine at position 324 on the biological behaviors of DENV-2 NS1 in relation with the literature (pages 8-9).

Reviewer 2 Report
Revision for Stability of Dengue 2 Non-structural glycoprotein 1 (NS1) is affected by the nature of basic residue at position NS1-324
In this manuscript submitted to Current Issues in Molecular Biology Ogire et al., describe the stability of the glycoprotein NS1 from DENV2 virus when mutating two amino acids. The authors have suggested that mutating these residues, especially K324) decreased stabilization of the protein and is associated with oxidative stress and pro-inflammatory cytokine activation
Overall, the introduction, method and conclusions are clear, concise, and appropriate with the current literature. The experiments within this paper are well executed and for most parts appropriately interpreted, though there are several issues and comments that need to be addressed prior to publication. These include:
Major points:
- Figure 1: The authors should include immunofluorescence images of transfected cells demonstrating transfection efficiency and signal intensity
- Figure 1 B: The authors did not include how the western blot signal intensity was calculated. There seems to be less intense B-actin in some wells. Was this considered when calculation NS1 signal intensity. Also, the authors should include the western blot ladder used to specify weight of the protein
- Figure 2: What was the MG132 solubilised in? Do the control samples and negative MG132 samples contain the solvent used? If not, this should be added to assess if the solvent concentration is having an effect on the cells and protein expression
- Figure 3 A: The relative increase from WT NS1 and K324 NS1 is approximately 0.05. Although this is a “significant” increase, is this relevant? Does this increase influence any downstream events? For example, an easy assessment would be to test by western blot weather this increase actives the integrated stress response by phosphorylating eIF2-alpha
- Has changes these amino acids changed the structure of the protein? I recommend that the authors should use Alphafold to predict the structure of NS1 and each mutant, and comment on how these mutations have changed the structure of the protein
Author Response
December 30th, 2022
To Current Issues in Molecular Biology (CIMB)
Section Biochemistry, Molecular and Cellular Biology
REF: Revised version of MS#cimb-2107494-R1 (Communication)
Dear Editors,
Dear Reviewers,
Thank you for your high consideration of our submitted manuscript.
Please in the revised document our point-to-point responses to Reviewers #1, #2, and #3.
Two major changes made to the article can be found below.
#1. In the revised version of the manuscript, we provided new Figures 1 and 6. A tridimensional structure prediction was performed on DENV-2 NS1 protein (Figure 1). A synthetic scheme on the biological behavior of DENV-2 NS1 mutant in Huh7 cells has been proposed as Figure 6.
#1. In the revised version of the manuscript, we provided new Figure S1. Immunofluorescence assay was performed on Huh7 cells expressing a recombinant DENV-2 NS1 protein.
We greatly appreciate the efforts of the three reviewers and the editorial team of CIMB to help us make this work the strongest possible, and we look forward to your reply.
Thank you very much for your assistance and we look forward to your earliest reply.
Sincerely yours,
Dr Marjolaine ROCHE
Corresponding author
Dr Marjolaine ROCHE, PhD
Faculty of Health Biology, La Reunion University
UM 134 PIMIT
CYROI platform, 97490 Sainte-Clotilde, La Réunion. France.
e.mail: marjolaine.roche@univ-reunion.fr
We greatly thank Reviewer #2 for his/her thoughtful comments and the opportunity to revise our manuscript.
Responses to Reviewer #2:
Q1. The authors should include immunofluorescence images of transfected cells demonstrating transfection efficiency and signal intensity.
We appreciate this point raised by the reviewer. A new supplementary Figure S1 has been added in the revised manuscript. According to the reviewer’s comment, the text has been modified in the revised manuscript (Results section, page 4 lines 146-55 and Supplementary data section, page 14, lines 446-54).
Q2. Figure 1 B: The authors did not include how the western blot signal intensity was calculated. There seems to be less intense ß-actin in some wells. Was this considered when calculation NS1 signal intensity. Also, the authors should include the western blot ladder used to specify weight of the protein.
We appreciate this point raised by the reviewer. In figure 1A, Western blot signal intensity was done with the Software Image J. The signal intensity of loading protein ß-actin has been found equivalent to all conditions. Relative abundance corresponds to the protein intensity measured for the DES-14 rNS1 mutants compared to wild-type rNS1. Using pre-stained natural protein standards for SDS-PAGE and immunoblotting, we have estimated the apparent molecular weight of each protein shown in Figures 2A and 3A. According to the reviewer’s comment, the text has been modified in the revised manuscript (Materials and Methods section, page 3, lines 90-92; Results section, page 5, lines 182-83; Results section, page 6, lines 211-12).
Q3. Figure 2: What was the MG132 solubilised in ? Do the control samples and negative MG132 samples contain the solvent used? If not, this should be added to assess if the solvent concentration is having an effect on the cells and protein expression.
Please accept our apologies for this omission. The peptide aldehyde MG132 has been solubilized in DMSO. For experiments, cells were incubated with 10 µM MG132 in 0.1% DMSO final concentration or 0.1% DMSO as vehicle control. DMSO at 0.1% final concentration caused no observable toxic effects to Huh7 cells. According to the reviewer’s comment, the text has been modified in the revised manuscript (Materials and Methods section, page 3, line 98-102).
Q4. Figure 3 A: The relative increase from WT NS1 and K324 NS1 is approximately 0.05. Although this is a “significant” increase, is this relevant? Does this increase influence any downstream events? For example, an easy assessment would be to test by western blot weather this increase actives the integrated stress response by phosphorylating eIF2-alpha.
We appreciate this point raised by the reviewer. Cellular ROS are known to be highly unstable and a very low concentration of ROS may a have a major effect on cell metabolism. Consequently, the relative weak increase of ROS detected in Huh7 cells expressing the NS1 mutant compared to wild-type NS1 is significant. We agree with the reviewer to investigate the downstream pathway of the oxidative stress activation by studying eIF2-alpha. Unfortunately, we did not do it but we keep this point in mind for later experiments. According to the reviewer’s comment, the text has been modified in the revised manuscript (Results section, page 6 ,lines 226-28).
Q5. Has changes these amino acids changed the structure of the protein? I recommend that the authors should use Alphafold to predict the structure of NS1 and each mutant, and comment on how these mutations have changed the structure of the protein.
We appreciate this point raised by the reviewer. A 3D structure prediction of DES-14 NS1 protein and mutants was performed using PHYRE2 protein fold recognition server. The predicted structures were analyzed with PyMOL 2.5, a program for interactive visualization of tridimensional proteins. Structural analysis of DES-14 NS1 showed that mutations NS1-R272K, NS1-R324K, or NS1-(R272K, R324K) have no obvious effect on DENV-2 NS1 conformation. According to the reviewer’s comment, a new Figure 1 has been added and the text has been modified in the revised manuscript (Results section, pages 3-4, lines 132-45).
/…/

Reviewer 3 Report
In this paper, Ogire et al have investigated the point mutational effect on the expression and stability of dengue viral ns1 protein. They have also shown the oxidative stress and pro-inflammatory cytokine activation in Huh7 cell expression to understand the associated defect due to mutation. It is an exciting piece of work since it is essential to study the structure-function relation of one of the most vital proteins in dengue. Although the manuscript is warranted publication, there are some major comments the authors should address.
1. For the broader readership of the journal CIMB, the authors should describe a little more (at least 2-4 sentences)about the effect of single point mutations on different proteins of pathogenic origins other than dengue and some relevant journals (e.g., https://doi.org/10.1038/s41467-020-19808-4; https://doi.org/10.1038/s41598-019-39185-3; https://doi.org/10.1021/acs.langmuir.8b00354; https://doi.org/10.1128/EC.00018-10;
https://doi.org/10.1021/acschembio.9b00327; https://doi.org/10.1021/acs.jpcb.6b11948) should be referenced.
2. The authors are recommended to provide the ns1 protein structure and the site of mutation in Figure 1
3. A schematic explaining the findings drawn by the authors should be provided.
4. What can be the physiochemical reason behind the alteration in function while arginine residue is replaced by lysine?
Author Response
December 30th, 2022
To Current Issues in Molecular Biology (CIMB)
Section Biochemistry, Molecular and Cellular Biology
REF: Revised version of MS#cimb-2107494-R1 (Communication)
Dear Editors,
Dear Reviewers,
Thank you for your high consideration of our submitted manuscript.
Please in the revised document our point-to-point responses to Reviewers #1, #2, and #3.
Two major changes made to the article can be found below.
#1. In the revised version of the manuscript, we provided new Figures 1 and 6. A tridimensional structure prediction was performed on DENV-2 NS1 protein (Figure 1). A synthetic scheme on the biological behavior of DENV-2 NS1 mutant in Huh7 cells has been proposed as Figure 6.
#1. In the revised version of the manuscript, we provided new Figure S1. Immunofluorescence assay was performed on Huh7 cells expressing a recombinant DENV-2 NS1 protein.
We greatly appreciate the efforts of the three reviewers and the editorial team of CIMB to help us make this work the strongest possible, and we look forward to your reply.
Thank you very much for your assistance and we look forward to your earliest reply.
Sincerely yours,
Dr Marjolaine ROCHE
Corresponding author
Dr Marjolaine ROCHE, PhD
Faculty of Health Biology, La Reunion University
UM 134 PIMIT
CYROI platform, 97490 Sainte-Clotilde, La Réunion. France.
e.mail: marjolaine.roche@univ-reunion.fr
We greatly thank Reviewer #3 for his/her thoughtful comments and the opportunity to revise our manuscript.
Responses to Reviewer #3:
Q1. For the broader readership of the journal CIMB, the authors should describe a little more (at least 2-4 sentences) about the effect of single point mutations on different proteins of pathogenic origins other than dengue and some relevant journals.
We appreciate this point raised by the reviewer. According to the reviewer’s comment, the text has been modified on Introduction section, page 2, line 47-49 and new references have been added in the revised manuscript.
Q2. The authors are recommended to provide the NS1 protein structure and the site of mutation in Figure 1.
We appreciate this point raised by the reviewer. A 3D structure prediction of DES-14 NS1 protein and mutants was performed using PHYRE2 protein fold recognition server)L The predicted structures were analyzed with PyMOL 2.5, a program for interactive visualization of tridimensional proteins. Structural analysis of NS1 showed that mutations NS1-R272K, NS1-R324K, or NS1-(R272K, R324K) have no obvious effect on DENV-2 NS1 conformation. According to the reviewer’s comment, a new Figure 1 has been added and the text has been modified in the revised manuscript (Results section, pages 3-4, lines 132-45).
Q3. A schematic explaining the findings drawn by the authors should be provided.
We appreciate this point raised by the reviewer. According to the reviewer’s comment, a new Figure 6 has been added and the text has been modified in the revised manuscript (Concluding remarks section, page 9, lines 296-99).
Q4. What can be the physicochemical reason behind the alteration in function while arginine is replaces by lysine ?
Although arginine and lysine are two basic residues, only the latter is considered as substrate for protein ubiquitination, SUMOylation, or acetylation. According to the reviewer’s comment, the text has been modified in the revised manuscript (Concluding remarks section, page 9, lines 301-02).

Round 2
Reviewer 2 Report
Dear Author's
Thank you for the detailed response to my comments. I wish you all the best.
Cheers
Reviewer 3 Report
The authors put effort into addressing all the points raised. The current version is sufficiently improved, and the manuscript is recommended for publication in its current form.
Best,